

# Systematic drug repositioning through mining adverse event data in ClinicalTrials.gov

Eric Wen Su and Todd M. Sanger

Advanced Analytics Hub, Eli Lilly and Company, Indianapolis, IN, United States of America

## ABSTRACT

Drug repositioning (i.e., drug repurposing) is the process of discovering new uses for marketed drugs. Historically, such discoveries were serendipitous. However, the rapid growth in electronic clinical data and text mining tools makes it feasible to systematically identify drugs with the potential to be repurposed. Described here is a novel method of drug repositioning by mining ClinicalTrials.gov. The text mining tools I2E (Linguamatics) and PolyAnalyst (Megaputer) were utilized. An I2E query extracts "Serious Adverse Events" (SAE) data from randomized trials in ClinicalTrials.gov. Through a statistical algorithm, a PolyAnalyst workflow ranks the drugs where the treatment arm has fewer predefined SAEs than the control arm, indicating that potentially the drug is reducing the level of SAE. Hypotheses could then be generated for the new use of these drugs based on the predefined SAE that is indicative of disease (for example, cancer).

## INTRODUCTION

Drug repositioning (i.e., drug repurposing) involves the identification and development of new uses for existing drugs (*Ashburn & Thor, 2004*). The best known example of drug repositioning is the serendipitous discovery of the additional use of thalidomide for the treatment of painful sores associated with leprosy. In 1964, Dr. Jacob Sheskin used thalidomide to help a patient sleep, unexpectedly, the thalidomide also healed the patient's sores and eliminated his pain (*Ashburn & Thor, 2004*; *Sheskin, 1965*). This discovery shows that clinical data could be the most direct and reliable source of drug repositioning.

However, systematic drug repositioning efforts since 1964 have not been based on clinical data. Typical approaches include high-throughput screening of marketed drugs (*Qosa et al., 2016*), targeted testing of a class of drugs for a new disease area (*Wu et al., 2016a*), and *in silico* methods (*Hodos et al., 2016*; *Mullen et al., 2016*), usually based on drug-target interactions (*Coelho, Arrais & Oliveira, 2016*; *Zheng et al., 2015*).

Described here is a novel approach to drug repositioning using data from randomized clinical trials. Text mining tools have been used to extract serious adverse event (SAE) data, identify drugs with fewer events related to diseases or associated symptoms in the drug arm than in the control arm, and rank the drugs based on the $z$-score of log odds ratio.

Corresponding author
Eric Wen Su, ewsu@lilly.com

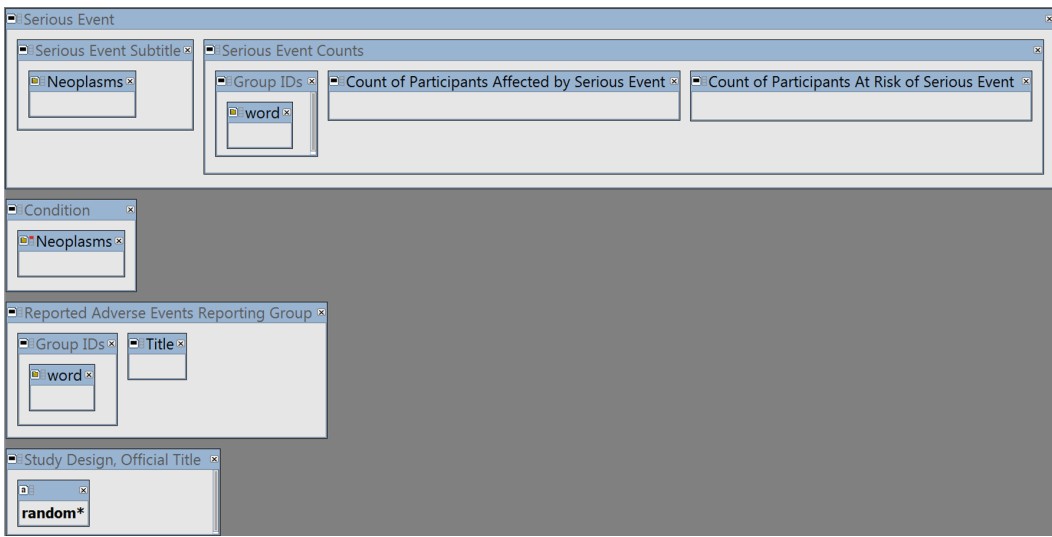

**Figure 1** **The I2E query.** See Supplemental Information 1 to reproduce the query by copying and pasting the YAML script into the I2E Pro interface. The query was run on the I2E index that covers the data posted in ClinicalTrials.gov up to August 14, 2016.

## MATERIALS & METHODS

A text mining query was developed to extract SAE data from clinical trial data posted at ClinicalTrials.gov. ClinicalTrials.gov is a registry of federally and privately funded clinical trials conducted in the United States and around the world, and contains rich biomedical data from over 220,000 studies in 191 countries. The query was built using Linguamatics' I2E, a literature text mining tool based on natural language processing and linguistic analytics (*Cormack et al., 2015*; *Galijatovic-Idrizbegovic et al., 2016*).

The query (shown in Fig. 1) has 4 main elements:

- To extract Serious Adverse Events classified as cancerous, the combined cancer terms and synonyms from MeSH (https://www.nlm.nih.gov/mesh/) and NCI (http://www.cancer.gov/research/resources/terminology) were loaded into the query region "Serious Event Subtitle" of ClinicalTrials.gov (the "Neoplasms" class).
- The same "Neoplasms" class was negated in the "Condition" region to exclude cancer trials.
- To link the SAE counts to the relevant study arm (i.e., drug or placebo etc.), the group (study arm) IDs and description ("Title") were extracted from the Reporting Groups region.
- The wildcard "random*" was required in the Study Design or Official Title region to ensure that only randomized trials are reported.

The Excel output from the I2E query in Fig. 1 was loaded into PolyAnalyst (Megaputer) for reformatting and calculating the odds ratios (OR) and *z*-score. The final table was sorted by *z*-score. PolyAnalyst is a commercial text mining tool. The specific tasks described here could also be accomplished by an open-source tool such as KNIME, R, or Python.

The formula for calculating odds ratio (*OR*), standard error (*SE*), 95% confidence interval lower and upper limits (*LowerLimit* and *UpperLimit*), and *z*-score are as follows:

Let $Cs$ = Number of patients with **S**AE in **C**ontrol arm; $Cn$ = **N**umber of patients in **C**ontrol arm and $Ds$ = **N**umber of patients with **S**AE in **D**rug arm; $Dn$ = **N**umber of patients in **D**rug arm.

$$OR = \frac{Ds/(Dn - Ds)}{Cs/(Cn - Cs)}$$

The distribution of log(*OR*) is approximately normal with:

$$SE = \sqrt{\frac{1}{Cs} + \frac{1}{Cn - Cs} + \frac{1}{Ds} + \frac{1}{Dn - Ds}}$$
$$LowerLimit = \exp(\log(OR) - 1.96SE)$$
$$UpperLimit = \exp(\log(OR) + 1.96SE)$$

The null hypothesis is that there is no difference between drug and control arm (expected mean OR = 1). Therefore,

$$z = \frac{\log(OR) - \log(1)}{SE} \text{ or } z = \log(OR)/SE$$

Since the $Cs$ and $Ds$ are usually small, SE, lower and upper limits, and *z*-score may not be meaningful for hypothesis testing. However, *z*-scores are still useful to rank drugs for hypothesis generation on drug repurposing.

Also because of the multiple comparison nature of the algorithm, the results should only be used for hypothesis generation, not for making any conclusions.

For drugs with *z*-scores $\leq -1.96$, we reviewed the biomedical literature on the drugs, the drug targets, and the disease pathways to see if the hypothesis is consistent with the current scientific knowledge. The literature review was performed using the text mining tool I2E (*Bandy, Milward & McQuay, 2009*).

## RESULTS

The I2E query in Fig. 1 was run on the ClinicalTrails.gov index updated on August 14, 2016. The report contains 105,399 SAE events classified as cancer, from 2,861 randomized trials. An example of the extracted data is shown in Fig. 2.

The I2E output table was reformatted as illustrated in Table 1 to have one row per trial per SAE (type of cancer).

If a row has less than 3 patients with SAE in the control arm, it is deleted. This is because the goal is to find drugs that have fewer cancer SAEs in the drug arm than in the control arm. After the deletions, the table has only 601 rows left.

If a row has 0 patients with SAE in the drug arm, the 0 value is replaced with 0.3. These replacements enable the ranking of the drugs that have no cancer SAE in the drug arm. Without the replacements, all such rows will have zero for OR and minus infinity for the *z*-score.

The final table with calculated columns is shown in Table 2. The drugs were ranked by sorting the *z*-score from the lowest value to the highest.

**A**

**Serious Adverse Events**

| | Daclizumab | Placebo |
|---|---|---|
| Basal cell carcinoma [†][1] | | |
| # participants affected / at risk | 4/216 (1.85%) | 3/207 (1.45%) |

**B**

| ClinicalTrials.gov ID | Serious Adverse Event | Study Arm | Number of Patients with SAE | Number of Patients |
|---|---|---|---|---|
| NCT00048165 | Basal cell carcinoma | Daclizumab | 4 | 216 |
| NCT00048165 | Basal cell carcinoma | Placebo | 3 | 207 |
| NCT00048581 | BASAL CELL CARCINOMA | Abatacept (ABA) | 1 | 258 |
| NCT00048581 | BASAL CELL CARCINOMA | Placebo (PLA) | 0 | 133 |
| NCT00089661 | Benign breast neoplasm | Denosumab 60 mg Q6M | 0 | 129 |
| NCT00089661 | Benign breast neoplasm | Placebo | 1 | 120 |
| NCT00089661 | Benign ovarian tumour | Denosumab 60 mg Q6M | 1 | 129 |
| NCT00089661 | Benign ovarian tumour | Placebo | 0 | 120 |

**Figure 2** **An example of the data extracted from ClinicalTrials.gov (A) into Excel (B) by the I2E query described above.** The top two rows in (B) show the data extracted from the table in (A). The precision of the I2E query described above is 100%, and the recall is estimated as 99% assuming 1% of the cancer terms that the trial sponsors used are not among the cancer synonyms collected by MeSH or NCI.

The results in Table 2 could range from false positive findings to possible signals for drug repositioning hypotheses. Therefore, we evaluated the drugs for cancer by other research from the current biomedical literature.

The V501 vaccine (Table 2, Row 1) arm had less cervical dysplasia events than the control in a clinical trial on the prevention of papillomavirus infection. Papillomavirus is already known to be associated with cervical dysplasia (*Firnhaber et al., 2009*), a precursor lesion of cancer of the cervix (*Kesic, Petkovic & Milacic, 1990*). We consider this top hit as a positive control that supports the credibility of our approach, since the prevention of the viral infection would naturally lead to the prevention of cervical dysplasia.

The data in Table 2, Row 2 suggest that Telmisartan might be useful to prevent colon cancer (note that Clopidogrel is in both the Drug and Control arm, so we did not investigate Clopidogrel further). Recent cell-based studies reported that Telmisartan exerts anti-tumor effects by activating peroxisome proliferator-activated receptor-$\gamma$ (*Li et al., 2014*; *Pu, Zhu & Kong, 2016*; *Wu et al., 2016b*). The algorithm presented here provides the first evidence
**Table 1  A sample of the reformatted table.**

| ClinicalTrials.gov ID | Serious adverse event | Number of patients with SAE in control arm | Number of patients in control arm | Control arm | Number of patients with SAE in drug arm | Number of patients in drug arm | Drug arm |
|---|---|---|---|---|---|---|---|
| NCT00089791 | Bladder cancer | 3 | 3,876 | Placebo | 4 | 3,886 | Denosumab 60 mg Q6M |
| NCT00089791 | Breast cancer | 25 | 3,876 | Placebo | 34 | 3,886 | Denosumab 60 mg Q6M |
| NCT00089791 | Colon cancer | 8 | 3,876 | Placebo | 11 | 3,886 | Denosumab 60 mg Q6M |
| NCT00120289 | Lung neoplasm malignant | 14 | 1,696 | Placebo + Simvastatin | 8 | 1,718 | ERN + Simvastatin |
| NCT00120289 | Malignant melanoma | 4 | 1,696 | Placebo + Simvastatin | 1 | 1,718 | ERN + Simvastatin |
| NCT00120289 | Non-small cell lung cancer | 4 | 1,696 | Placebo + Simvastatin | 0.3 | 1,718 | ERN + Simvastatin |
| NCT00143507 | Colon cancer | 7 | 5,430 | Placebo | 5 | 5,477 | Ivabradine |
| NCT00143507 | Rectal cancer | 6 | 5,430 | Placebo | 3 | 5,477 | Ivabradine |

**Table 2  The final table with calculated columns.** The rows are sorted by z-score. Only the top 6 rows are shown (see Supplemental Information 2 for all 162 rows with $z < -1$).

| Drug | Serious adverse event | Ds | Dn | Cs | Cn | Control | SE | OR | Lower limit | Upper limit | z | Clinical Trials.gov ID |
|---|---|---|---|---|---|---|---|---|---|---|---|---|
| V501 | Cervical dysplasia | 20 | 480 | 46 | 468 | Placebo | 0.28 | 0.40 | 0.23 | 0.69 | −3.33 | NCT00378560 |
| Clopidogrel/ Telmisartan | Colon cancer | 4 | 5,000 | 14 | 5,023 | Clopidogrel/ Placebo | 0.57 | 0.29 | 0.09 | 0.87 | −2.20 | NCT00153062 |
| Vorapaxar | RECTAL CANCER | 4 | 13,186 | 13 | 13,166 | Placebo | 0.57 | 0.31 | 0.10 | 0.94 | −2.06 | NCT00526474 |
| Phylloquinone | Cancer | 3 | 217 | 11 | 223 | Placebo | 0.66 | 0.27 | 0.07 | 0.98 | −1.99 | NCT00150969 |
| Clopidogrel + ASA | Pancreatic carcinoma | 1 | 3,772 | 8 | 3,782 | Placebo + ASA | 1.06 | 0.13 | 0.02 | 1.00 | −1.96 | NCT00249873 |
| Core- phase: Aliskiren | Gastric cancer | 1 | 4,272 | 8 | 4,285 | Core- phase: Placebo | 1.06 | 0.13 | 0.02 | 1.00 | −1.96 | NCT00549757 |

Notes.

$Ds$, Number of patients with SAE in Drug arm; $Dn$, Number of patients in Drug arm; $Cs$, Number of patients with SAE in Control arm; $Cn$, Number of patients in Control arm. The original indications of the trials were (from top to bottom): HPV Infections, Stroke, Atherosclerosis, Osteoporosis, Atrial Fibrillation, and Type 2 Diabetes.

from a randomized clinical trial indicating that Telmisartan may be viable as a repurposed prevention for colon cancer.

Phylloquinone (Table 2, Row 4) is a vitamin (vitamin K1) supplement rather than a prescription drug. K vitamins + sorafenib induce apoptosis in human pancreatic cancer cell lines (*Wei, Wang & Carr, 2010*). A prospective cohort analysis found that individuals who

increased their intake of dietary phylloquinone might have a lower risk of cancer than those who did not (*Juanola-Falgarona et al., 2014*). The data from the randomized trial in Table 2 suggest that vitamin K1 might actually help prevent cancer ($OR = 0.27$, 95% CI [0.07–0.98]). The potential cancer prevention by vitamin K1 is especially intriguing because one can get more than 1,000% daily value of vitamin K1 by simply eating one cup of cooked kale or spinach (https://www.healthaliciousness.com/articles/food-sources-of-vitamin-k.php).

The clinical trial in Table 2, row 6, tested Aliskiren for cardiovascular and renal disease in patients with type 2 diabetes. The SAE data from this study show that only 1 out of 4,272 patients in the Aliskiren arm reported gastric cancer versus 8 out of 4,285 patients in the placebo arm. A recent paper described that Aliskiren inhibits renal carcinoma cell lines proliferation *in vitro* (*Hu et al., 2015*). The data from this randomized clinical trial suggest the possible repurposing of Aliskiren for cancer.

Lastly, our literature search found no direct link between Vorapaxar (Table 2, Row 3) or Clopidogrel (Table 2, Row 5) and cancer prevention or treatment. Thus, these data in Table 2 could be the first sign that Vorapaxar or Clopidogrel might be useful for cancer or could be interpreted as false positive findings since we have made no attempt to adjust the multiplicity (multiple comparisons) in this exploratory analysis.

Above are only six outputs from our repositioning algorithm for one type of disease. The method described here could be used to identify other candidates for repositioning on any diseases that are reported as serious adverse events in ClinicalTrials.gov.

## DISCUSSION

Presented here is a novel drug repositioning method that reveals potential new uses of existing drugs directly from clinical trial data. This article provides only a rudimentary way to conduct drug repositioning using text mining tools on ClinicalTrials.gov. However, it could serve to stimulate other investigational initiatives to use clinical data to repurpose drugs, supplements, or even food to help prevent or treat diseases.

Serious adverse event data from randomized trials in the ClinicalTrials.gov were used because randomized trials are controlled experiments. However, ClinicalTrials.gov is only a tiny part of clinical data that could lead to the discovery of new use of existing drugs. Electronic medical record databases have much more clinical data than ClinicalTrials.gov. Other large sources of clinical data include the Federal Adverse Event Reporting System and social media (*Nugent, Plachouras & Leidner, 2016*). These data could provide new information not only on marketed drugs, but also on supplements and food.

Computational drug repositioning usually involves the vast genome data and sophisticated machine learning techniques (*Li et al., 2016*). In contrast, the work described here uses relatively small clinical trial data on ClinicalTrials.gov, which has been proved useful in other works to identify combination therapy (*Wu et al., 2015*) and pharmacogenomics information (*Li & Lu, 2012*). The algorithm presented here is simple and direct. Combining this work with text mining (*Tari & Patel, 2014*) may lead to better methodologies for drug repurposing.

Compared to traditional drug development, repositioned drugs have the advantage of decreased development time and costs given that significant toxicology and safety data will

have already been accumulated, drastically reducing the risk of attrition during the drug discovery and development process.

## CONCLUSIONS

The rapidly growing clinical data could be extracted and analyzed for drug repositioning utilizing text mining tools. Repositioning non-cancer drugs with low toxicity or even vitamin supplements for cancer might provide tangible benefits for patients.

The method described could be used for drug repositioning not only for cancer but also for other diseases and symptoms reported as adverse events. It might help other investigators to develop better ways to utilize the fast growing data in ClinicalTrials.com to reposition drugs for unmet medical needs.

The work we described here could merely help identify possible new uses of existing drugs to be investigated further. Prospective clinical trials would be required to provide the necessary evidence to have such new uses approved by regulatory agencies.

## ACKNOWLEDGEMENTS

The authors are grateful for the statistical review by Dr. Margaret Gamalo, the expert editing by Drs. Marjo Gazak and Jane Reed, and insightful advice and review by Dr. Stephen Ruberg.

### Funding
The authors received no funding for this work.

### Competing Interests
Eric Wen Su and Todd M. Sanger are employees of Eli Lilly and Company, United States of America.

### Author Contributions
- Eric Wen Su conceived and designed the experiments, performed the experiments, analyzed the data, contributed reagents/materials/analysis tools, wrote the paper, prepared figures and/or tables, reviewed drafts of the paper.
- Todd M. Sanger reviewed drafts of the paper, supervised research and contributed to the Discussion section.

### Data Availability
The raw data has been supplied as a Supplementary File.

### Supplemental Information
Supplemental information for this article can be found online at http://dx.doi.org/10.7717/peerj.3154#supplemental-information.

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
