# Peer review of "Systematic drug repositioning through mining adverse event data in ClinicalTrials.gov"

_PeerJ, doi:10.7717/peerj.3154_

## Round 0.1 · original submission · Major Revisions

The reviewers carried out a detailed analysis of the manuscript. Both reviewers made several suggestions that will improve the overall quality of the manuscript.

Reviewer 1 ·

Basic reporting

Lack of literature references. See more in General comments.

Experimental design

See more in General comments.

Validity of the findings

See more in General comments.

Additional comments

This manuscript describes an automated approach for drug repurposing, i.e. identifying new usage for existing drugs. The rationale for the approach is to utilize clinical trials data to realize drugs that have statistically significant fewer patients with reports of serious adverse effects in drug arm compared to the control arm. Drugs that satisfy such criteria are hypothesized to be a potential treatment for the serious adverse effects. In this manuscript, the authors focused on finding serious adverse effects related to cancer. The authors highlighted the top-6 novel drug indications and discussed the literature findings that may support the new indications. 

While the approach seems to be novel, the manuscript at its current form is not yet ready for journal publication. It lacks the discussion of related work that uses computational approaches to identify new drug indications. At the minimum, the authors need to discuss and compare their work with the following related work: 
Li, Jiao, et al. "A survey of current trends in computational drug repositioning." Briefings in bioinformatics 17.1 (2016): 2-12. 
Wu M, Sirota M, Butte AJ, Chen B. Characteristics of drug combination therapy in oncology by analyzing clinical trial data on clinicaltrials.gov. Pac Symp Biocomput 2015;20 68–79. 
Li J, Lu Z. Systematic identification of pharmacogenomics information from clinical trials. 
Tari LB, Patel JH. Systematic drug repurposing through text mining. Methods Mol Biol 2014;1159:253–67. 

The approach at its state seems to be "half-baked". While the authors showed that the approach identified some potential novel drug indications, the authors had to manually perform literature search to see if the suggested drug indication has any scientific findings for support. This can be a cumbersome task and for it to be a comprehensive approach, the authors should extend its current state of the approach so that the algorithm can identify scientific literature support for drug indications. The ones that don't may indicate truly novel drug indications. 

Other minor points: 
Need to have an introduction of Linguamatics I2E, PolyAnalyst. The authors should discuss if this approach would work without these commercial tools. 
Need to show the corresponding extracted results. Show, for example, how the row for NCT00048165 in Table 1 was extracted from the original clinical trial page. What is the precision and recall for the results extracted by I2E? 
Need to specify the dates covered by the index, since not everyone has access to I2E. 
Should show original indication in Table 3.

·

Basic reporting

The article describes a new drug repositioning method using a text-mining approach of clinical data to identify drugs that could be repurposed. The identification of “Serious Adverse Events”, then analysis using statistical approaches examines the potential of the approach for identifying repurposing candidates. The article is well written, overall, with only a small number of grammatical errors. These have been marked up in the PDF for review and addressing as necessary. The introduction and background regarding the purpose of the work is clear and well-defined. My review of the structure of the article appears to be in compliance with the standards for PeerJ (based on my personal limited experience with the journal). The research as described is clear and the single figure is relevant and appropriate. The raw data is provided as an Excel spreadsheet according to PeerJ policy.

Experimental design

The method as described should be reproducible.

Validity of the findings

The research study as described is to the best of my knowledge novel and seems to be within the scope of the journal. The conclusion is VERY SHORT and could likely be extended to more fully encapsulate the work that was performed and the future promise of the work.

Additional comments

I find this research to be well conducted and well written but encourage expansion of the conclusion. I would suggest moving the last few lines of the discussion to the conclusion to help identify the future benefits of this work

---

## Round 0.2 · accepted · Accept

The revised manuscript improved a great deal and it can be accepted as it is.

Reviewer 1 ·

Basic reporting

No comment

Experimental design

No comment

Validity of the findings

No comment

Additional comments

The authors have addressed the concerns I raised in my previous review, and thank you for that. The only minor suggestion I have is to remove Line 55-56 that states "We were not able to perform the data extraction without this commercial tool.". While the authors may not be able to use other tools, that does not mean no other tools or future tools can perform such a function. So I recommend removing this statement.

·

Basic reporting

No comment. I have nothing to add following my first review.

Experimental design

No comment. I have nothing to add following my first review.

Validity of the findings

No comment. I have nothing to add following my first review.

Additional comments

I acknowledge that the authors have accepted the feedback of both of the reviewers and made the appropriate adjustments to the manuscript and feel that the manuscript has improved in both quality and depth. Specifically, inclusion of the additional references, as recommended by reviewer 1 and then additionally by the authors, helps the reader in navigating to other relevant works. I still recommend that this article be accepted for publication and feel that it has improved since my first review.